# MicroRNA-323-5p Involved in Dexmedetomidine Preconditioning Impart Neuroprotection

**DOI:** 10.3390/medicina59091518

**Published:** 2023-08-23

**Authors:** Hyunyoung Seong, Daun Jeong, Eung Hwi Kim, Kyung Seob Yoon, Donghyun Na, Seung Zhoo Yoon, Jang Eun Cho

**Affiliations:** 1Department of Anesthesiology and Pain Medicine, Anam Hospital, Korea University College of Medicine, Seoul 02841, Republic of Korea; hello406@naver.com (H.S.); seob1009@naver.com (K.S.Y.); bmssdh7942@gmail.com (D.N.);; 2Institute for Healthcare Service Innovation, Korea University, Seoul 02841, Republic of Korea; neutrino93@gmail.com (D.J.); b612m45@gmail.com (E.H.K.)

**Keywords:** dexmedetomidine, hypoxia–ischemia, brain, ischemic preconditioning, miRNAs, neuroprotection, therapeutic uses

## Abstract

*Background and Objectives*: Cerebral ischemia is one of the major preoperative complications. Dexmedetomidine is a well-known sedative–hypnotic agent that has potential organ-protective effects. We examine the miRNAs associated with preconditioning effects of dexmedetomidine in cerebral ischemia. *Materials and Methods*: Transient infarcts were induced in mice via reperfusion after temporary occlusion of one side of the middle cerebral artery. A subset of these mice was exposed to dexmedetomidine prior to cerebral infarction and miRNA profiling of the whole brain was performed. We administered dexmedetomidine and miRNA-323-5p mimic/inhibitor to oxygen–glucose deprivation/reoxygenation astrocytes. Additionally, we administered miR-323-5p mimic and inhibitor to mice via intracerebroventricular injection 2 h prior to induction of middle cerebral artery occlusion. *Results*: The infarct volume was significantly lower in the dexmedetomidine-preconditioned mice. Analysis of brain samples revealed an increased expression of five miRNAs and decreased expression of three miRNAs in the dexmedetomidine-pretreated group. The viability of cells significantly increased and expression of miR-323-5p was attenuated in the dexmedetomidine-treated oxygen–glucose deprivation/reoxygenation groups. Transfection with anti-miR-323-5p contributed to increased astrocyte viability. When miRNA-323-5p was injected intraventricularly, infarct volume was significantly reduced when preconditioned with the miR-323-5p inhibitor compared with mimic and negative control. *Conclusions*: Dexmedetomidine has a protective effect against transient neuronal ischemia–reperfusion injury and eight specific miRNAs were profiled. Also, miRNA-323-5p downregulation has a cell protective effect under ischemic conditions both in vivo and in vitro. Our findings suggest the potential of the miR-323-5p inhibitor as a therapeutic agent against cerebral infarction.

## 1. Introduction

Cerebral ischemia is a major perioperative complication, and the perioperative stroke ranges from 0.1 to 1.9% according to the underlying risk factors [1,2]. The lack of effective neuroprotective therapies makes treatment difficult. Even after successful recanalization, subsequent ischemia/reperfusion (I/R) injury can cause irreversible cerebral damage.

Dexmedetomidine, a selective α2-agonist, has sedative, analgesic, and anxiolytic effects and is widely used in various anesthetic fields, such as intraoperative sedation in the operating room and postoperative management in the intensive care unit (ICU). Compared to other sedatives, DEX has several benefits for neurosurgery, such as minimal respiratory depression, lesser interference with neuronal function, and airway patency maintenance during awake procedures [3].

Moreover, DEX is a promising alternative as a neuroprotective agent. Multiple studies have reported that DEX exerts protective effects against ischemic injuries in the brain, heart, and kidneys [4]. In cerebral ischemia/reperfusion injuries, pretreatment with DEX imparts neuroprotective effects by regulating cell apoptosis, proliferation, and inflammatory pathways [5]. Also, DEX helps maintain the balance of microregional oxygen supply and consumption in the brain, preserves immune function, and improves cognitive function [6]. Despite their neuroprotective effects, the underlying molecular mechanisms remain largely unknown. The possible mechanisms related to DEX preconditioning include the modulation of brain-derived neurotrophic factor in astrocytes [7] and suppression of Toll-like receptor 4 in the nuclear factor κB pathway [8].

“Preconditioning” is a well-known phenomenon wherein small amounts of injurious stimuli can confer resistance against subsequent harmful events. In cerebral ischemia, several preconditioning methods, such as inhalation of volatile anesthetics, short-term ischemia, hypothermia, and immune activation, exert remarkable protective effects [9]. This phenomenon occurs through an endogenous cascade pathway and induces tolerance to ischemic injury [10].

During the last decade, research on the modulation of microRNA (miRNA) in the neuroprotective effect of DEX has been of interest to many researchers [11]. Essentially, miRNAs are single-stranded RNA molecules composed of 21–23 nucleotides that regulate gene expression at the post-transcriptional level via targeted mRNA degradation or translation suppression. These miRNAs are known to play important roles in the pathogenesis of cerebral ischemia [12] as well as preconditioning-induced ischemic tolerance [13]. Besides, they have been shown to play a role as potential diagnostic or therapeutic agents in cerebral ischemia [14]. Therefore, we aimed to profile specific miRNAs involved in dexmedetomidine preconditioning that confer neuroprotection and explore the potential therapeutic effects of one of these miRNAs.

Of the profiled miRNAs, miR-323-5p was selected to conduct further research on. Few studies have shown that miR-323-5p controls apoptosis in cerebral infarction conditions by modulating the transforming growth factor-β1 (TGF-β1)/Smad3 signaling pathway [15] or BRI3 [16], but no studies have been conducted regarding the mechanisms associated with DEX.

## 2. Materials and Methods

### 2.1. DEX-Preconditioning Induced Neuroprotection and Associated miRNA Profiling

#### 2.1.1. Animal Grouping and Transient Cerebral Ischemia Model

Male C57BL/6 mice provided by Orient Bio (Seongnam, Republic of Korea) were raised at the semi-specific pathogen-free facility under a 12 h light/dark cycle. We used 8-week-old mice weighing approximately 20 g. Thereafter, we randomly divided the mice into three groups: (1) the sham group, that underwent incision and closure; (2) the middle cerebral artery occlusion (MCAO) group, and (3) the MCAO + DEX group. Cerebral infarction/reperfusion models were established in mice by MCAO. Dexmedetomidine hydrochloride (PRECEDEX^®^, Hospira, Lake Forest, IL, USA) was diluted in sterile normal saline.

Before transient MCAO, all mice were anesthetized with isoflurane (CAS #. 26675-46-7, Hana Pharm. Seoul, Republic of Korea). We used the Koizumi method to induce MCAO, as described in previous studies [17,18]. After making a midline incision in the neck, the right carotid artery was meticulously separated, and 6-0 silicon-coated nylon filament was inserted until it encountered a slight increase in resistance, which signified the origin of the middle cerebral artery. After one hour of occlusion, the filament was gently removed to allow for reperfusion. In the MCAO + DEX group, DEX was intraperitoneally injected 30 min before MCAO, and the maximal safe dose of DEX (100 µg/kg, intraperitoneal) was used according to previous studies [19,20]. The same volume of sterile normal saline was intraperitoneally injected as a vehicle into the MCAO group. A flow diagram summarizing the process is shown in Figure 1A.

#### 2.1.2. Neurological Deficit Scoring and Triphenyl Tetrazolium Chloride (TTC) Staining

The neurological deficits were scored by a researcher who was blinded to the experimental groups. After the mice were subjected to 24 h of MCAO, the neurological function of each mouse was scored using modified Neurological Severity Scores (mNSS) as previously reported [21]. A neurological deficit score of 2 to 3 was used as the success criterion for establishing the MCAO model. The mice that did not meet this criterion were excluded, along with those that underwent subarachnoid hemorrhaging or death within 24 h.

Following, all the mice were euthanized with sevoflurane (CAS #. 28523-86-6; Hana Pharm, Seoul, Republic of Korea) after 24 h of reperfusion, and their brains were harvested. The isolated brains were placed on an ice-cold metal plate and sectioned into four coronal slices of 2 mm thickness for TTC (CAS #. 298-96-4, Sigma-Aldrich, St. Louis, MO, USA) staining. After 1 h, the stained brain tissues were fixed with 4% paraformaldehyde (Biosesang, Seongnam, Republic of Korea). The posterior surface of each slice was photographed under a digital camera and the size of the infarct area was analyzed by Image J software (Rawak Software Inc., Stuttgart, Germany). The total infarct volume was calculated as the sum of the infarct areas of the four sections and is presented as a percentage of the volume of the left hemisphere.

#### 2.1.3. Isolation of miRNA

Total RNA was extracted using TRI Reagent^®^ (Merck Darmstadt, Germany) from whole brain tissue. The extracted RNA was eluted in 100 µL diethylpyrocarbonate-treated water (Gendepot, Katy, TX, USA). Using NanoDrop One (Thermo Fisher Scientific, Waltham, MA, USA), ODs at 260 and 280 nm were determined, and samples with OD260/280 between 1.7 and 2.1 were selected for subsequent processing. The extracted RNA was profiled using Affymetrix miRNA 4.0. This was outsourced to Macrogen (Seoul, Republic of Korea).

#### 2.1.4. Raw Data Preparation and Statistical Analyses

Total RNA (1 μg from each sample) was labeled and hybridized using the FlashTag Biotin HSR RNA Labeling Kit (Genisphere LLC, Hatfield, PA, USA). Poly(A) tailing, and ligation were performed using the same kits. Biotin-labeled RNA was hybridized at 45 °C for 16–18 h in a GeneChip hybridization oven. After washing and staining, the GeneChips were scanned using a GeneChip scanner. The extracted data were analyzed using robust multiarray analysis detection above the background normalization method. The array export data were normalized and log transformed. The miRNAs with a 2-fold or greater difference in expression were analyzed, and each group sample was compared using an independent *t*-test under the null hypothesis that there was no difference between them. The *p*-value was adjusted to account for false discovery rate. Hierarchical clusters were analyzed in the gene set that showed significant fold-change values.

### 2.2. Neuroprotection Is Induced by miR-323-5p In Vitro

#### 2.2.1. Cellular Oxygen-Glucose Deprivation (OGD)/Reoxygenation Model

The human astrocyte cell line U-373MG was obtained from the Korean Cell Line Bank (KCLB). The cells were cultured according to the protocol provided by KCLB. Supplemented RPMI 1640 medium without oxygen and glucose was used as OGD medium. To remove the remaining oxygen in the OGD medium, the medium was placed in a hypoxic modular incubator chamber (StemCell Technologies, Vancouver, Canada) for at least 2 h before the experiment. A modular incubator chamber was used according to the manufacturer’s instructions. Cells were seeded in culture plates and grown until 80–90% confluency was reached. After the growth medium was removed, the cells were washed with phosphate-buffered saline and degassed OGD medium was added to the wells. Then, the gas in the chamber was exchanged to maintain hypoxic conditions (basal atmosphere: 95% N_2_, 5% CO_2_, 1% O_2_ at 37 °C). Cellular OGD was performed by placing the culture plates in a modular hypoxic incubator chamber. In the OGD condition, cells were exposed to the OGD medium for one hour and reoxygenated. When reoxygenation began, the medium was changed to DEX-free RPMI 1640 growth medium, and cells were subjected to normoxic conditions at 37 °C in 5% CO_2_ containing atmosphere. A flow diagram summarizing the process was shown in Figure 1B.

#### 2.2.2. Dexmedetomidine Application to the Cellular OGD Model

DEX was dissolved in dimethyl sulfoxide (DMSO, Biosesang, Yongin, Republic of Korea) and added to U-373MG cells at a final concentration of 10 μM in growth media and OGD medium. The dose of DEX was determined according to previous studies [22,23]. DEX was applied to cells 30 min before and during OGD. The maximum concentration of DMSO in the medium was 0.1%.

#### 2.2.3. Cell Viability Assay

Cell viability was determined using a 3-(4,5-dimethlythiazol-2-yl)-2,5-diphenyltetrazolium bromide (MTT; Sigma-Aldrich, CAS #. 298-93-1) proliferation assay. Briefly, MTT was dissolved in sterile PBS (5 mg/mL) and diluted in serum-free RPMI 1640 at a ratio of 1:9 (*v*/*v*). The 0.5 mg/mL MTT solution was added and incubated for an hour at 37 °C. Subsequently, the supernatants were discarded and DMSO was added to the wells to dissolve the formazan crystals. Solutions containing dissolved formazan crystals were transferred into a 96-well plate and the absorbance was measured at 570 nm using an i30 spectrophotometric microplate reader (Molecular Bioscience, San Jose, CA, USA). Cell viability was determined as the percentage or level of absorbance relative to that of the normoxic control.

#### 2.2.4. Transfection of miR-323-5p In Vitro

MiRNA sequences were determined by referring to miRTarBase [24] (http://www.mirtarbase.cuhk.edu.cn, assessed on 23 June 2022) web servers. The mature miRNA sequence of mice miR-323-5p (mmu-miR-323-5p) is AGGUGGUCCGUGGCGCGUUCGC. The mimic and inhibitor of miR-323-5p were obtained from Ambion (Cat # 4464066-mimic and Cat # 4464084-inhibitor, Waltham, MA, USA). For miRNA transfection, we seeded cells at a density of 3 × 10^5^ cells/well in six-well plates and they were cultured until 30–50% confluency was reached. The culture medium was changed to Opti-MEM (reduced serum medium) before transfection to increase the transfection efficacy. We mixed 0.4 nmol miR-323-5p with 15 µL Lipofectamine^®^ 3000 reagent (Thermo Fisher Scientific Waltham, MA, USA) and transfected the cells according to the manufacturer’s instructions. After transfection, the cells were cultured under hypoxic conditions (basal atmosphere: 95% N_2_, 5% CO_2_, 1% O_2_ at 37 °C) in OGD medium for one hour and reoxygenated, as described previously [22]. When reoxygenation began, cells were exposed to normoxic conditions (5% CO_2_ incubator at 37 °C) for 6 h and then the fresh medium was added to the cells. Thereafter, the cells were incubated for the next 48 h. To confirm cell viability and the expression of miR-323-5p, we performed MTT assay and quantitative RT-PCR (qRT-PCR), respectively. qRT-PCR analysis of miR-323-5p was performed using MystiCq^®^ microRNA qPCR Assay Primer (Sigma-Aldrich).

#### 2.2.5. Quantitative Real-Time PCR (qRT-PCR)

Total RNA was extracted using the TRIzol reagent (Sigma-Aldrich) and chloroform (Merck, Darmstadt, Germany). For reverse transcription, the Mir-X miRNA First-Strand Synthesis Kit (Takara, Kusatsu, Japan) was used according to the manufacturer’s instructions. The expression of miR-323-5p was determined using the Mir-X miRNA qRT-PCR TB Green^®^ Kit (Takara) and specific primers (miR-323-5p qRT-PCR primer from Merck and housekeeping U6 primer from Takara). qRT-PCR was performed using a CFX96 Real-Time PCR Detection System (Bio-Rad, Hercules, CA, USA). The expression rate of miR-323-5p was normalized to that of U6 housekeeping miRNA (Takara, Kusatsu, Japan) using the Delta-Delta Ct (ddCt) method to determine the relative expression.

### 2.3. Intracerebroventicular Injection of miR-323-5p In Vivo

Two hours prior to MCAO, the mmu-miR-323-5p mimic and negative control tagged with green fluorescent protein were administered via intracerebroventricular injection (ICV) at a concentration of 0.8 nmol dissolved in 4 μL PBS. The mice were anesthetized and placed in a stereotactic head frame (RWD Life Science, Shenzhen, China) while lying face down. A midline scalp incision was made, and a burr hole was drilled on the right side of the skull, 0.2 mm posterior and 0.8 mm lateral to the bregma. Using a glass pipette injector attached to a Micro4 micro syringe pump (World Precision Instruments, Sarasota, FL, USA), the miR-323-5p mimic or inhibitor (4 μL) was microinfused into the right lateral ventricles at a depth of 2.3 mm. The needle was left in place for an additional 5 min to prevent possible leakage and was slowly withdrawn within 4 min. After removing the needle, the incision was sutured, and the mice were allowed to recover. To check whether miRNAs spread properly in the brain, brains were sliced into 1 mm slices 2 h after intraventricular injection and checked with a fluorescence microscope.

### 2.4. Statistical Analysis

Prior to conducting the statistical analyses, a normality test was conducted using the Shapiro–Wilk test in order to confirm that the data follow normal distribution. The infarct size and neurobehavior score were analyzed by the non-parametric Kruskal–Wallis test, followed by the Dunn’s multiple comparisons test. In order to confirm cell viability from hypoxia time or miR-323-5p preconditioning, one-way analysis of variance (ANOVA) test followed by the Tukey’s multiple comparisons test was conducted. In order to confirm cell viability between the DEX-treated and untreated group as well as miR-323-5p expression, two-way ANOVA followed by Bonferroni’s multiple comparisons test was performed. All statistical analyses were performed using GraphPad Prism 8.0 (GraphPad Software, Inc., San Diego, CA, USA). The differences were statistically significant at *p* < 0.05.

## 3. Results

### 3.1. DEX Preconditioning Induced Neuroprotection and Related miRNA Profiling

#### 3.1.1. Effect of DEX on Cerebral Infarct Volume in Mice

In total, 83 mice were randomly divided into sham, MCAO, and MCAO + DEX groups (3 for sham, 40 for MCAO, and 40 for MCAO + DEX). The mortality rate was 18% (7/40) in the MCAO group, and 13% (5/40) in the MCAO + DEX group, and thus a total of 33 mice in the MCAO group and 35 mice in the MCAO + DEX group were analyzed. No differences in age or weight were observed between the groups (Table 1). After the transient MCAO, cerebral infarcts were observed in the cortex. Compared to the MCAO group, the DEX pre-treatment group effectively reduced the infarct volume and improved the neurological scores (Figure 2A).

#### 3.1.2. Effect of DEX on miRNA Profiling in the Ischemic Brain

A total of 36,249 miRNAs were extracted as raw data and processed for expression analysis. The miRNAs were filtered by species, and flag-preprocessed data (3076 miRNAs) and filtered data (680 miRNAs) were obtained. The statistical test cutoff level was |fold-change| ≥ 2.0, and *p*-value < 0.05.

The expression of the following five miRNAs was significantly upregulated in the DEX group (Table 2): miR-34c-5p, miR-448-3p, miR-6953-5p, miR-6956-5p, and miR-7052-5p. Conversely, the expression of three miRNAs (miR-188-5p, miR-323-5p, and miR-6984-5p) was significantly downregulated (Table 3). The miRNA expression has been represented as a volcano plot and hierarchical clustering is shown in Figure 3 and Figure 4.

### 3.2. Neuroprotection Is Induced by miR-323-5p In Vitro

#### 3.2.1. Effect of DEX in Cellular OGD/Reoxygenation Model

Cell viability was measured using the MTT proliferation assay (Figure 5). As shown in Figure 5A, under OGD conditions, cell viability decreased in a time-dependent manner (*n* = 6, 1 h: 64.42 ± 5.92%, 2 h: 57.17 ± 4.64%, and 4 h: 48.35 ± 8.61% vs. normoxic control, *p*-value < 0.001 at each time point). As shown in Figure 5B, cell viability significantly increased in DEX (+) groups under OGD conditions compared to the DEX (−) groups during 6 to 24 h of reoxygenation (*n* = 3, DEX (−); 6 h: 31.81 ± 0.8%, 12 h: 41.26 ± 3.04%, and 24 h: 57.49 ± 3.4%. DEX (+); 6 h: 52.31 ± 4.33%, 12 h: 57.5 ± 3.05%, and 24 h: 72.11 ± 2.78%. *p*-value < 0.001, DEX (+) groups vs. DEX (−) groups at each time point).

#### 3.2.2. Expression of miR-323-5p in Cellular OGD/Reoxygenation Model

The expression of miR-323-5p in the cellular OGD/reoxygenation model and the DEX preconditioning effect were confirmed by qRT-PCR during 6 to 24 h of reoxygenation. Compared to the control group, the fold-change in the expression of miR-323-5p was significantly increased in the OGD/reoxygenation groups (Figure 5C, *n* = 3, fold-change; 6 h: 10.18 ± 2.82 and 24 h: 16.74 ± 4.8, *p*-value = 0.03 vs. normoxic control, at each time point.). In contrast, the expression levels of miR-323-5p were significantly attenuated in the DEX-treated OGD/reoxygenation groups (*n* = 3, fold-change; 6 h: 1.15 ± 0.53 and 24 h: 0.01 ± 0.01, *p*-value = 0.02 vs. OGD/reoxygenation group at each time point).

#### 3.2.3. The Inhibition of miR-323-5p Is Involved in OGD-Induced Apoptosis

To confirm the direct effect of miR-323-5p on cell survival under OGD conditions, we transfected astrocytes with the miR-323-5p mimic and inhibitor before OGD/reoxygenation. As shown in Figure 6A, cells transfected with miR-323-5p inhibitor before OGD/reoxygenation exposure showed significantly increased survival rate, compared to the control group (*n* = 6, control group; 100%, miR-323-5p inhibitor group; 224.3 ± 15.5%, miR-323-5p mimic; 166.7 ± 9.24%; *p*-value < 0.001, control group vs. miR-323-5p inhibitor group). Therefore, miR-323-5p downregulation improves astrocyte survival under OGD conditions. Quantitative PCR was used to demonstrate that the expression levels of miR-323-5p and miR-323-5p inhibitors were effectively downregulated (Figure 6B, *n* = 6, fold-change; miR-323-5p inhibitor: 0.09 ± 0.02 and miR-323-5p mimic: 0.71 ± 0.04. *p*-value < 0.001, control group vs. miR-323-5p). In contrast, miR-323-5p upregulation decreased cell survival after OGD. However, these results were not statistically significant.

### 3.3. Neuroprotection Is Induced by miR-323-5p In Vivo

In Figure 7, we demonstrate the effect of miR-323-5p on cerebral infarction. In total, 27 mice were randomly divided into sham, MCAO, and MCAO + 323-5p negative control (NC), MCAO + miR-323-5p inhibitor, and MCAO + miR-323-5p mimic groups. As shown in Figure 7C, administration of the miR-323-5p inhibitor effectively reduced the infarct volume (12.04 ± 0.67), compared to MCAO (18.31 ± 0.69, *p*-value < 0.001), NC (18.34 ± 1.42, *p*-value = 0.002), and mimic (16.04 ± 1.32, *p*-value = 0.02). Pre-injection of the miR-323-5p mimic did not reduce infarct volume compared with the NC and MCAO groups (*p*-value = 0.28 vs. NC, *p*-value = 0.24 vs. MCAO).

## 4. Discussion

In this study, we demonstrated that DEX preconditioning effectively decreased cerebral infarct size and profiled the miRNAs associated with neuroprotection. To the best of our knowledge, this is the first study to profile miRNAs involved in DEX-induced preconditioning effects. We identified five significantly upregulated and three significantly downregulated miRNAs during DEX preconditioning that conferred neuroprotection. We also observed that the suppression of one of the eight miRNAs, miRNA-323-5p, led to a protective effect against ischemic/reperfusion injury in both in vivo and in vitro experiments.

Perioperative cerebral infarction is one of the most undesirable outcomes for patients, surgeons, and anesthesiologists. The prevalence of perioperative cerebral infarction can be nearly as high as 10% in high-risk cardiac or neurosurgeries, and these patients have an 8-fold higher mortality rate than normal patients [25]. Treatment of acute cerebral ischemia still relies on recanalization strategies that have a limited therapeutic window. However, even after the blood flow is restored, post-reperfusion injury, including neuronal cell apoptosis or cerebral edema, can be induced. Recent studies have extensively focused on novel treatment strategies that can modify a series of pathological cascades. Despite this effort, the most promising agents in preclinical trials have failed to show clear benefits in clinical situations [26].

From this point of view, DEX is purportedly a promising neuroprotective agent based on both preclinical and clinical evidence. Animal research has shown that the administration of DEX after focal cerebral infarction results in an improvement in the size of the cortical infarction and an increase in the survival of cells [3]. In rats with global ischemia, preconditioning or post-conditioning with DEX reduced neuronal loss and improved neurofunction [3]. DEX has shown neuroprotection when used in human surgeries such as brain tumors and traumatic brain injury as an anesthetic [27]. It has a favorable profile with respect to the length of ICU stays, the risk of delirium, and long-term brain function in critically ill patients and effectively reduces inflammatory mediators and free radicals [28,29]. Clinically, DEX serves as both an anesthetic adjuvant and a sedative for patients in the ICU who are critically ill and in need of mechanical ventilation. Most of these patients have a high risk of cerebral ischemia.

Consistent with the findings from previous studies [3,4,11], our findings revealed that DEX preconditioning attenuated brain infarct size in transient ischemic mice as well as astrocyte apoptosis during OGD. These findings thus suggest the feasibility of the development of clinical drugs based on the neuroprotective action of DEX.

Recent studies have extensively focused on novel treatment strategies that can modify a series of pathological cascades of cerebral infarction. Experimental studies have shown that miRNA modulation is involved in the neuroprotective effects of DEX. The miRNAs have also been found to play an important role in the pathogenesis and preconditioning of cerebral ischemia [11]. Therefore, they are expected to be novel therapeutic targets. The most attractive characteristic of miRNA-based therapy is that a single miRNA can target multiple genes. Thus, it is useful in situations wherein multiple genes are deregulated such as cerebral ischemia. In clinical practice, miRNA-based treatment has been used for the treatment of hepatitis C virus or acute myocardial ischemia [30,31].

Our findings show that both in vivo and in vitro inhibition of miRNA-323-5p is involved in DEX preconditioning-induced neuroprotection. Our results indicate the possibility of administering a single miRNA as a therapeutic agent for cerebral infarction in clinical settings. To demonstrate the effects of miR-323-5p on OGD/reoxygenation in vitro, we generated a cellular OGD model using astrocytes. First, we found cell survival decreased in the OGD/reoxygenation model in a time-dependent manner. Consistently, we found that miR-323-5p expression was upregulated in the OGD/reoxygenation model and increased several folds over time. Furthermore, the downregulation of miR-323-5p expression attenuated apoptosis and provided protection against OGD/reoxygenation. Consistent with the results of previous studies, our results showed that miR-323 expression was upregulated under I/R conditions in vitro. The precise mechanism of the neuroprotective effect of miRNA-323 remains unclear, but it may regulate apoptosis during cerebral infarction by targeting the TGF β1/SMAD3 signaling pathway [15] or by modulating BRI3 [16], which is involved in I/R injury-induced neuronal apoptosis.

Next, we conducted another in vivo study to confirm whether the miRNA has a potential protective effect in an animal model of cerebral ischemia. When the miR-323-5p inhibitor was injected two hours before MCAO induction, we found that cerebral infarct size was reduced. The fact that modulation of a single gene has a cell-protective effect both in vivo and in vitro indicates that miR-323-5p may play an important role in the pathophysiology of DEX-protective roles.

Research on miRNA changes in DEX-related neuroprotective effect has been of interest to many researchers, and DEX has shown promising potential as a neuroprotective agent over the past few years [11]. There are, however, several differences between ours and previous studies. First, previous studies [32,33,34,35,36] did not analyze the whole brain but rather focused on a single gene and its target protein; these studies thus did not reflect diverse gene interactions in a clinical situation. We believe that whole-brain profiling is more accurate in identifying complex gene interactions, and our approach is useful in detecting changes in miRNAs in real situations by analyzing multiple genes simultaneously through profiling.

Moreover, most studies have focused on the post-conditioning effect of DEX by administering it immediately before reperfusion or ischemia. In most clinical situations, cerebral infarction occurs when DEX is administered during neurosurgery or ICU care, making it more appropriate to investigate preconditioning. As little is known about the miRNAs related to the preconditioning effect of DEX, we expect that our miRNA profiling will provide new insights into the development of protective and therapeutic agents against cerebral infarction in high-risk patients.

Third, most studies did not determine whether specific miRNAs are effective in vivo or in vitro. In contrast, our study showed that miRNA-323-5p was downregulated both in vivo and in vitro under ischemic/reperfusion conditions. In addition, when preconditioned with miR-323-5p inhibitor, low ischemic region in vivo and high cell viability in vitro were shown, which means that miR-323-5p may have a neuroprotective role.

Lastly, since the impact of miRNA-mediated gene expression and cellular functions varies depending on the type of cell, we administered miR-323-5p to astrocytes to verify whether the miRNA acts on the cells that were investigated, while most studies have focused on neuronal cells. Astrocytes are the most abundant cell type in the central nervous system and are essential for regulating normal brain functions. These cells are also being studied as promising therapeutic targets because of their important role in neurological recovery after stroke [37,38]. Therefore, we believe that our research will serve as a foundation for future studies on astroglial therapeutic targets in patients with infarction.

The remaining seven miRNAs identified in our profiling study may provide new directions for research on miRNAs in cerebral protection. Eight miRNAs found to be related to the DEX-mediated preconditioning effect were miR-34c-5p, miR-448-3p, miR-6953-5p, miR-6956-5p, miR-7052-5p, miR-188-5p, miR-323-5p, and miR-6984-5p. This trend was consistent with the findings of several previous studies [39,40,41,42]. Among them, miR-34-5p was upregulated during DEX preconditioning in our study. In other studies, there was a notable decrease in the expression of miR-34-5p in vivo and in vitro ischemia models. Its overexpression reportedly decreased brain infarct and edema size and improved functional scores in MCAO rats [40]. Accordingly, miR-34c-5p appears to play a pivotal role in stroke by controlling the expression of inflammatory cytokines and proteins involved in apoptotic signaling pathways.

In traumatic spinal cord injury, mmu-miR-6953-5p expression is downregulated and upregulated after DEX preconditioning. According to the mechanism proposed by Ying et al., mmu-miR-6953-5p is the site where circRNA.7079 binds and regulates the expression of Lgal3. The authors suggested that miR-6953-5p might be related to the circRNA-miRNA-mRNA-ceRNA network that mediates apoptosis [39].

The miR-188-5p expression was upregulated during OGD, and the knockout of miR-188-5p improved cell viability, which was also found to be downregulated in DEX preconditioning. The possible mechanism by which miRNA-188-5p acts is to regulate neuronal apoptosis and survival rate by inhibiting phosphatase and tensin homologs deleted on chromosome 10 [42].

A limitation of this study is that although cerebral infarction was induced by a limited area of MCAO, miRNAs were profiled from the whole brain. Therefore, bias may arise when comparing the miRNA profiles in the MCAO brain and MCAO + DEX brain.

## 5. Conclusions

We showed that DEX preconditioning exerts neuroprotective effects both in vivo and in vitro. Using in vivo miRNA profiling, we identified eight miRNAs involved in DEX preconditioning. This study demonstrated that miR-323-5p downregulation plays a role in DEX preconditioning-induced neuroprotection.

## Figures and Tables

**Figure 1 medicina-59-01518-f001:**
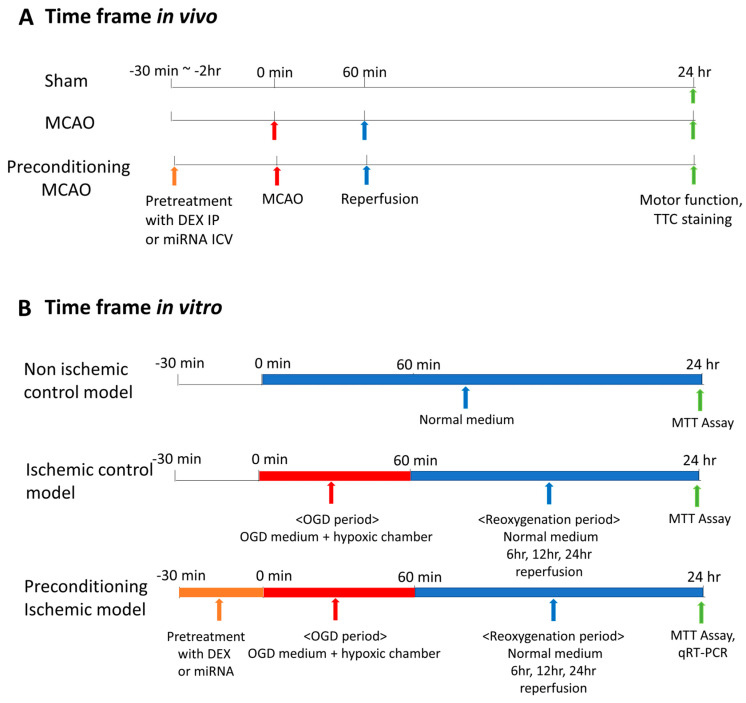
Flow diagram of in vivo and in vitro study. (**A**) In vivo study. To confirm the preconditioning effect of DEX, DEX was injected intraperitoneally 30 min before induction of MCAO. To confirm the preconditioning effect of miRNA-323-5p, miR-323-5p negative control, mimic, and inhibitor was injected into the intracerebral ventricle 2 h before MCAO induction. Reperfusion was performed 60 min after MCAO induction, and after conducting the neurologic score test 24 h later, the brains were taken out and subjected to TTC staining. (**B**) In vitro study. Before exposure to the OGD period, DEX or miRNA-323-5p mimic/inhibitor was pretreated for 30 min, and cell viability was evaluated 24 h later. MCAO, middle cerebral artery occlusion; IP, intraperitoneal; ICV, intracerebroventricular injection; TTC, triphenyl tetrazolium chloride; MTT, 3-(4,5-dimethlythiazol-2-yl)-2,5-diphenyltetrazolium bromide; OGD, oxygen–glucose deprived.

**Figure 2 medicina-59-01518-f002:**
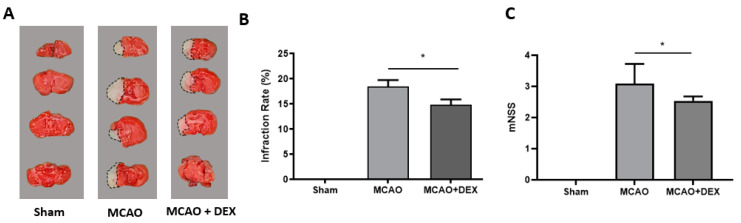
Effect of dexmedetomidine (100 µg intraperitoneal administration) on cerebral infarct volume and modified neurological severity score ((**A**) TTC staining of infarct sections; (**B**) Percentage of infarct volume; (**C**) mNSS) after 24 h of reperfusion in mice challenged with MCAO-induced transient ischemia. *n* = 3 Sham, 33 MCAO, 35 DEX + MCAO group, (* *p*-value *<* 0.05). DEX, dexmedetomidine; MCAO, middle cerebral artery occlusion; TTC, triphenyl tetrazolium chloride; mNSS, modified neurological severity score. Data are expressed as the mean ± standard deviation.

**Figure 3 medicina-59-01518-f003:**
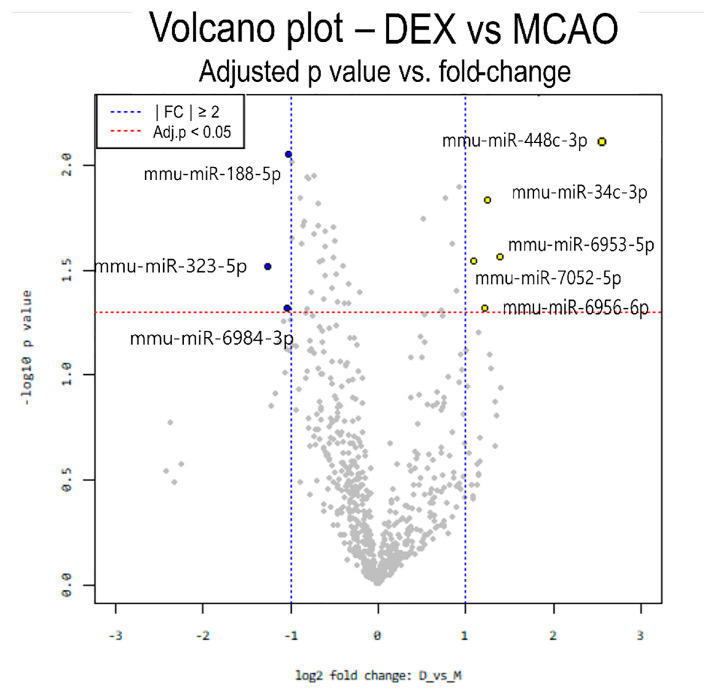
Differential expression analysis for miRNAs involved in DEX-mediated preconditioning of ischemic brain tissues is presented using a volcano plot. The *X*-axis represents log-fold-change and the *Y*-axis is −log10-adjusted *p*-value. The red dotted line indicates a 0.05 significance level, whereas the blue dotted line indicates |fold-change| ≥ 2.

**Figure 4 medicina-59-01518-f004:**
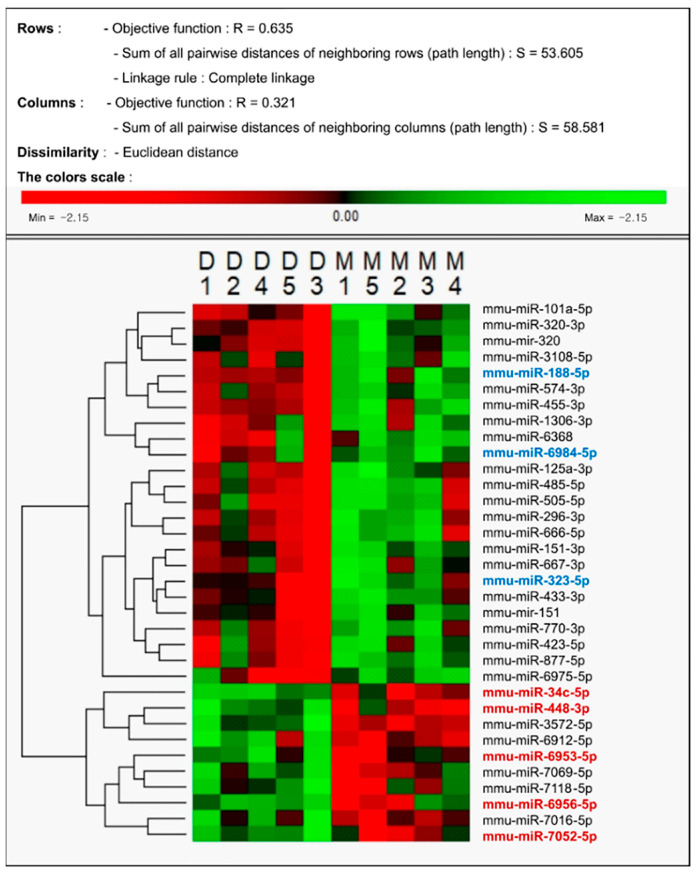
Hierarchical clustering of differentially expressed microRNAs (miRNA) in DEX-preconditioned ischemic brain tissues. Blue bold miRNAs denote the decreased miRNAs, and red bold miRNAs denote the increased miRNAs in this study. The heat map shows all differentially expressed miRNAs across the MCAO vs. MCAO + DEX groups. D, MCAO + DEX group; M, MCAO group.

**Figure 5 medicina-59-01518-f005:**
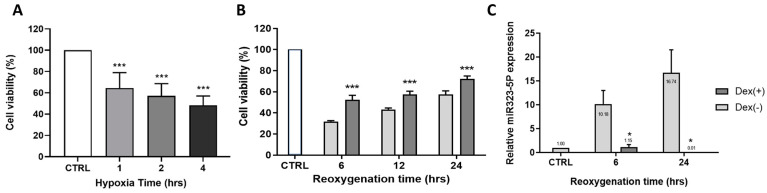
Cell viability and effect of DEX in OGD condition. (**A**) In a time-dependent manner, cell viability was decreased (***, *p*-value < 0.001 vs. non-ischemic cells). (**B**) In OGD/reoxygenation model, cell viability was increased in DEX (+) group compared to the DEX (−) group during 6 to 24 h of reoxygenation (***, *p*-value < 0.001 vs. DEX (−) cells). (**C**) The expression rate of miR-323-5p in OGD/reoxygenation condition. In DEX (+) group, miR-323-5p fold-change attenuated significantly compared to DEX (−) group. (*, *p*-value < 0.05 vs. DEX (−) cells).

**Figure 6 medicina-59-01518-f006:**
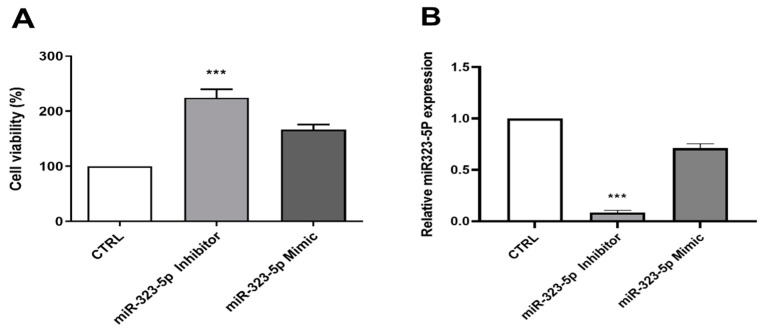
Effect of miR-323-5p inhibitor and mimic on cell viability. (**A**) Cell viability significantly increased when miR-323-5p inhibitor was transfected before OGD/reoxygenation compared to the control group. (**B**) The transfection was confirmed with quantitative PCR. (***, *p*-value < 0.001 vs. control).

**Figure 7 medicina-59-01518-f007:**
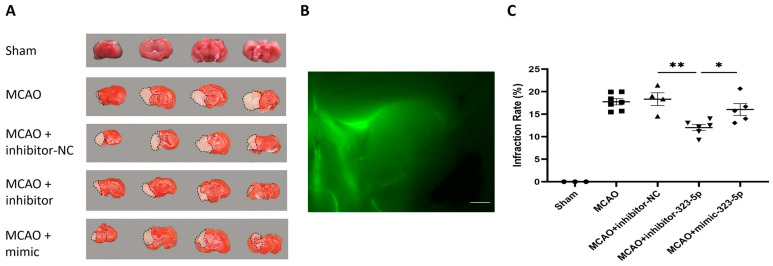
Administration of miR-323-5p inhibitor reduced the infarction. (**A**) Brain slices stained with TTC in different groups at 24 h post MCAO are shown as representative images. (**B**) Brain-slice image taken 2 h after injection of GFP-labeled miR-323-5p into the Rt ventricle. Injected miRNA spreads to the ipsilateral brain. Scale bar = 0.8 mm. (**C**) Quantitative analysis of infarction volume in different groups. Data are expressed as mean ± standard deviation. ●: sham, ■: MCAO, ▲: MCAO + inhibitor-NC, ▼: MCAO + inhibitor-323-5p, ◆: MCAO + mimic. *n* = 3 sham, 6 MCAO, 6 MCAO + inhibitor-NC, 6 MCAO + inhibitor-323-5p, 6 MCAO + mimic (*, *p*-value < 0.05, **, *p*-value < 0.01). DEX, dexmedetomidine; MCAO, middle cerebral artery occlusion; TTC, triphenyltetrazolium chloride.

**Table 1 medicina-59-01518-t001:** Characteristics of mice.

	Sham	MCAO	MCAO + DEX
Body weight (g)	21.3 ± 0.7	21.9 ± 0.3	21.9 ± 0.4
Age in weeks	8	8	8

DEX, dexmedetomidine; MCAO, middle cerebral artery occlusion.

**Table 2 medicina-59-01518-t002:** The miRNAs upregulated in the DEX group as compared to the MCAO group.

miRNA	MCAO	MCAO + DEX	Fold-Change	Adjusted *p*-Value
mmu-miR-34c-5p	7.17 ± 0.72	8.42 ± 0.30	2.37	0.015
mmu-miR-448-3p	2.59 ± 1.09	5.32 ± 1.19	6.63	0.006
mmu-miR-6953-5p	5.50 ± 0.92	6.89 ± 0.64	2.63	0.027
mmu-miR-6956-5p	2.63 ± 0.98	3.84 ± 0.30	2.32	0.048
mmu-miR-7052-5p	3.52 ± 0.75	4.62 ± 0.47	2.14	0.028

Data are expressed as the mean ± standard deviation. DEX, dexmedetomidine; MCAO, middle cerebral artery occlusion.

**Table 3 medicina-59-01518-t003:** The miRNAs downregulated in the DEX group as compared to the MCAO group.

miRNA	MCAO	MCAO + DEX	Fold-Change	Adjusted *p*-Value
mmu-miR-323-5p	7.26 ± 0.70	6.01 ± 0.80	2.38	0.031
mmu-miR-6984-5p	7.16 ± 0.44	6.12 ± 0.83	2.06	0.048
mmu-miR-188-5p	4.78 ± 0.49	3.75 ± 0.43	2.03	0.009

Data are expressed as the mean ± standard deviation. DEX, dexmedetomidine; MCAO, middle cerebral artery occlusion; FC, fold-change.

## Data Availability

The datasets generated during and/or the current study are not publicly available but are available from the corresponding author on reasonable request.

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
