# Peer review of "MicroRNA-323-5p Involved in Dexmedetomidine Preconditioning Impart Neuroprotection"

_medicina, 2023, doi:10.3390/medicina59091518_

Round 1
Reviewer 1 Report
This manuscript by Hyunyoung Seong et al, aimed to profile specific miRNAs involved in dexmedetomidine (DEX) preconditioning that confer neuroprotection and explore the potential therapeutic effects of miRNA-323-5p in a transient cerebral ischemia model. The manuscript is interesting, however, to be published in this journal the following demands are required:
Major concerns:
Summary and introduction
1. The focus of cerebral ischemia is limited to complications of neurosurgery. I suggest it is also associated with the cerebrovascular event (the main cause of ischemia).
2. The use of transfection with miRNA-323-5p is not clear
Introduction
3. The molecular mechanisms through which DEX exerts its neuroprotective effects are not clear. Detail.
4. The authors must associate and justify using miR-323-5p.
Methods
5. In a flow diagram, outline the number of animals used, per group and experiments and times. It is essential that this be detailed. Also include in vitro assays in the diagram.
6. It is necessary to incorporate the control group: sham or healthy + DEX.
7. Specify the thickness of the tissue sections and how many anatomical levels were considered to determine the infarcted volume. As well as the region of interest considered in the analysis.
8. The authors mention that they extracted total RNA from the whole brain. This is a bias since the model is of unilateral involvement and "delimited" anatomical territory.
9. Specify the statistical tests used for each experimental trial in a section.
10. It is necessary to confirm the motor condition in this model with MNSS.
Results
11. Statistically is it valid to compare 3 individuals vs 33?
12. In Fig 1A and 7A it is not the same anatomical level in the 3 conditions. Authors must illustrate at least 3 anatomical levels per condition.
13. The authors mention that both DEX and miRNA-323-5p have neuroprotective effects. However, it is not enough to only determine cell viability and TTC staining to assert the neuroprotective effect. The authors must at least determine apoptosis in the model, given the involvement of miRNA-323-5p in this death mechanism.
Minor concerns:
1. In the title include cerebral infarction and the use of miRNA-323-5p3.
2. Specify the percentage of patients with neurosurgery who present cerebral ischemia complications.
3. Add the DEX catalog number in the methods and not in the introduction
4. Verify the use of abbreviations throughout the manuscript.
5. Specify the DEX vehicle in vivo assays.
6. The supplementary material was not included.
7. Specify the % mortality of the model
8. At the bottom of Figure 1, specify the dose of DEX.
9. Improve the quality and size of all figures.
10. At the figure caption of Figure 3 define D and M.
11. In figure captions, specify the number of animals used.
12. Change the color code to differentiate Dex (-) vs CTRL
13. Add calibration bars where appropriate
14. Check the significant difference in Figure 3 c with respect to what is mentioned in the text
15. Missing to place the miRNA sequence
16. Verify what is described in L408-409.
17. Update your references since about 45% are before 2016.
Minor editing of English language required
Author Response
We appreciate the reviewers’ valuable revisions, which have helped us to considerably improve the manuscript and enhance its clarity. Please see below our point-by-point response to the reviewer’s comments. In addition, we have provided a revised version of the manuscript, with the changes marked in red. We hope that the changes incorporated in the revised manuscript satisfactorily address the reviewers’ concerns.

Reviewer 2 Report
The authors presents their work in the fiel of fundamental research adressing the MicroRNAs involvment in in dexmedetomidine preconditioning. Their findings showed the potential role of the miR-323-5p inhibitor as a possible neuroprotector in cerebral ischemia. The results are sound and the paper is well-written. I recommend it for publication.
There are some small issues, so that the authors should once have the manuscript checked by a native speaker
Author Response
Thank you very much for your review. We will work hard for further work.
Round 2
Reviewer 1 Report
I congratulate the authors for their excellent work.
Minor editing of English language required